# Origami-inspired soft fluidic actuation for minimally invasive large-area electrocorticography

Lawrence Coles [1,2], Domenico Ventrella [3], Alejandro Carnicer-Lombarte [1], Alberto Elmi[3], Joe G. Troughton [1,2], Massimo Mariello [2], Salim El Hadwe[1,4], Ben J. Woodington[1], Maria L. Bacci [3], George G. Malliaras [1], Damiano G. Barone [1,4] & Christopher M. Proctor [1,2] ✉

Electrocorticography is an established neural interfacing technique wherein an array of electrodes enables large-area recording from the cortical surface. Electrocorticography is commonly used for seizure mapping however the implantation of large-area electrocorticography arrays is a highly invasive procedure, requiring a craniotomy larger than the implant area to place the device. In this work, flexible thin-film electrode arrays are combined with concepts from soft robotics, to realize a large-area electrocorticography device that can change shape via integrated fluidic actuators. We show that the 32-electrode device can be packaged using origami-inspired folding into a compressed state and implanted through a small burr-hole craniotomy, then expanded on the surface of the brain for large-area cortical coverage. The implantation, expansion, and recording functionality of the device is confirmed in-vitro and in porcine in-vivo models. The integration of shape actuation into neural implants provides a clinically viable pathway to realize large-area neural interfaces via minimally invasive surgical techniques.

Electrocorticography (ECoG) implants are neural interfaces that consist of an array of electrodes to record brain activity from the surface of the brain. This type of implant placed either epi- or subdurally on the cortical surface, facilitates high spatiotemporal resolution electrophysiology over a large area of the brain[1]. The integrated electrodes in the ECoG array measure averaged local field potentials produced by neurons in the cerebral cortex through direct contact with its surface[2]. ECoG arrays are currently in clinical use as diagnostic tools for epilepsy treatment[3]. Once implanted, the ECoG array is used to map the cortex and identify the foci of the epileptic seizure, guiding surgical intervention[4]. ECoG devices have also been used in brain-computer interfaces[5,6] including applications for speech decoding[7,8], control of prosthesis[9–11] and closed-loop seizure control[12,13].

The capacity to monitor brain activity over centimetre length scales is critical for most ECoG applications. However, achieving such large area coverage typically requires cutting a window in the skull that is larger than the implanted ECoG array. This craniotomy procedure is highly invasive, increasing the risk of infection, surgical complications, and potential cosmesis issues for the patient[14]. A hospital stay is also required during cortical recording using large-area ECoG arrays, further increasing clinical costs and preventing the translation of these devices for chronic monitoring or prosthesis control[15]. Recent research has shown ECoGs can be made from softer[16,17] and thinner[18–22] materials to increase biocompatibility for a chronically implanted system, however, the invasiveness of implantation remains a key limitation to the wider adoption of these technologies.

[1]Department of Engineering, University of Cambridge, Cambridge, UK. [2]Institute of Biomedical Engineering, Engineering Science Department, University of Oxford, Oxford, UK. [3]Department of Veterinary Medical Sciences, Alma Mater Studiorum, University of Bologna, Ozzano dell'Emilia, Bologna, Italy. [4]Department of Clinical Neurosciences, University of Cambridge, Cambridge, UK. ✉e-mail: christopher.proctor@eng.ox.ac.uk

While large-area ECoG arrays are still used for the identification of epileptic foci, there is a clinical trend towards less invasive implants to reduce surgical risk and cost. Narrow cortical electrode strips can be implanted using burr-hole craniotomy by sliding the device in place subdurally, removing the requirement for a full craniotomy[23], with novel thin-film ECoG designs being able to slide onto the cortex with small, slit incisions in the skull[24]. Similarly, thin flexible mesh electrode arrays can be packaged within a microcatheter to allow introduction into the body via an injection[25–28]. However, the utility of such implants is limited as cortical coverage is restricted by the width of the incision in the skull, therefore large craniotomies could still be required for large-area cortical coverage. Compared to large ECoGs, brain penetrating Stereoelectroencephalography (SEEG) electrodes implanted through small burr-hole craniotomies reduce the surgical footprint and provide information at depth[29,30] at the expense of reduced cortical coverage and increased foreign body reaction due to penetration into brain tissue[31,32]. To overcome the coverage issues, multiple-depth electrodes are often implanted at different locations through several small burr holes which in turn increases the invasiveness. An approach that combines the reduced surgical footprint of SEEG and strip-ECoG devices, with the large spatial coverage of standard ECoG devices would ideally balance the competing clinical needs.

Concepts from soft robotics have recently converged with bioelectronic implants to create implants that can change shape and adapt within the body to enable minimally-invasive interfacing[33,34] or provide mechanical stimulation[35]. Through shape actuation, novel soft robotic implants have been demonstrated to mechanically match and conform to surrounding tissue to electrically map the internal chambers within the heart[36,37], to reduce the foreign body reaction of drug delivery devices[38,39] and enable implantation of neurostimulation spinal cord devices with minimally invasive surgical techniques[40]. Recently, there has also been a demonstration of using soft robotics to deploy an ECoG strip through a standard burr-hole craniotomy using eversion to deploy an initially inverted device[41], showing there is a clear clinical and engineering push for minimally invasive devices.

Here, we combine soft robotics with bioelectronics to demonstrate a minimally invasive subdural large-area ECoG implant (MI-ECoG). Thin-film, biocompatible polymers, commonly used in the fabrication of bioelectronic implants, are engineered to enable shape actuation through the integration of a fluidic chamber in the centre of the implant. The device can be folded for subdural implantation onto the surface of the cortex through a burr-hole craniotomy. Shape actuation enables large-area ECoG coverage previously only achieved with a full craniotomy via a surgical implantation analogue to existing clinical ECoG strip electrodes and SEEG implants. In addition to enabling less invasive ECoG, the presented innovations in implant and fluidic design may enable further development of novel shape-actuated implants that target mechanically challenging areas, expanding into complex brain regions such as within cortical sulci.

## Results

### Design and fabrication of the MI-ECoG

The MI-ECoG is a large, soft neural interface that can be folded into a small form factor, allowing insertion subdurally onto the cortical surface through a burr-hole craniotomy. Through the integration of a fluidic actuation chamber in the centre of the device, the device can be inflated to drive the unfolding of the device in situ thereby covering a large cortical region and enabling neural recordings (Fig. 1A). Thin-film fabrication techniques were used in creating the MI-ECoG device, including soft- and photolithography, to form a device made from silicone, parylene-C, gold, and poly(3,4-ethylenedioxythiophene) polystyrene sulfonate (PEDOT:PSS) (Fig. 1B).

For the fluidic platform of the MI-ECoG, we developed a unique device stack-up and polymer laser welding technique to achieve a device capable of sustaining the pressures required for shape actuation underneath the skull. In the centre of the device, a fluidic chamber is formed from two layers of parylene-C separated with a thin soap-based anti-adhesion layer. Using a $CO_2$ laser cutter allows the outline of the device to be defined, as well as columns in the centre of the chamber that act as tie-points to reduce vertical expansion during actuation. By tuning the parameters of the laser, bonding between the two layers of parylene-C is achieved, with this bonding enabling the fluidic chamber to contain high fluidic pressures capable of actuating the device. This chamber is integrated with monolithic silicone fabrication to further increase pressure tolerance, through the encapsulation of the parylene-C fluidic chamber with two layers of Polydimethylsiloxane (PDMS). The bonding between the two silicone layers that encapsulate the parylene-C chamber allows the formation of a 60–70 μm thick device, with the PDMS encapsulation strengthening resistance to leakage during actuation (full fabrication steps shown in Supplementary Fig. S1).

Visibility of successful device expansion is restricted given that the skull is still in place, therefore, intraoperative X-ray imaging is required to enable surgical tracking. Opaque X-ray markers made from a silicone-bismuth composite are integrated on each side of the device to track expansion. We have shown previously that these are flexible and biocompatible, suitable for integration into thin-film implants[42]. The MI-ECoG device presented includes a 32-electrode array consisting of 1.6 mm diameter PEDOT:PSS electrodes (Fig. 1B), with a combination PEDOT:PSS/Au tracks in order to minimise electrode impedance. The full cross-sectional structure of the MI-ECoG is shown in Fig. 1C, D. Alternative electrode designs that could be integrated with the fluidic component of the MI-ECoG are shown in the Supplementary Materials (Fig. S2).

In order to deploy the ECoG device in the subdural space without a full craniotomy, the device must be able to expand within the confined subdural space. The necessity of opening within the limited space guided the packaging approach. Rolling the implant, which has been shown previously to allow expansion in the spinal cord[40], prevents expansion due to the pressure from the brain trapping the device. Instead, by employing origami-inspired packaging, each side of the device was folded in a concertina pattern, with the width of each fold set to 1 mm using a spacer. By folding the sides of the device, the fluidic actuation drives lateral expansion that can overcome both the friction between the device and the arachnoid and the pressures required to slightly depress the brain and expand within the virtual subdural space.

For subdural deployment, the device is folded and placed with a custom introduction tool (Fig. 1E). Heat-shrink tubing was shaped around surgical forceps using a guide to create a rectangular tip with an inner dimension of 4 × 1.5 mm. This secures the folds of the device during implantation and helps maintain correct device orientation.

### In-vitro mechanical validation

In-vitro mechanical validation of the devices was required, given the challenge of device expansion in the limited subdural space. The maximum pressure that these devices could withstand was investigated (Fig. 2A). The devices were inflated at 1 ml/min with either air or water using a syringe pump coupled with a pressure transducer. The pressure was measured until failure, where a fluidic leak causes a large pressure drop. When inflated using air, the devices were able to hold 34–37 kPa of pressure, whilst inflation with water significantly increased the maximum pressure of the devices to 77–89 kPa. This maximal pressure tolerance is significantly higher than our previously reported shape-changing implants, with the laser welded parylene-C fluidic chamber enabling higher pressures to be used during expansion[40].

The expansion of the MI-ECoG device was tested in-vitro using a hydrogel brain/dura model to identify a suitable packaging methodology, and the fluidic pressures required for reliable expansion. To represent the mechanical characteristics of the brain, a composite

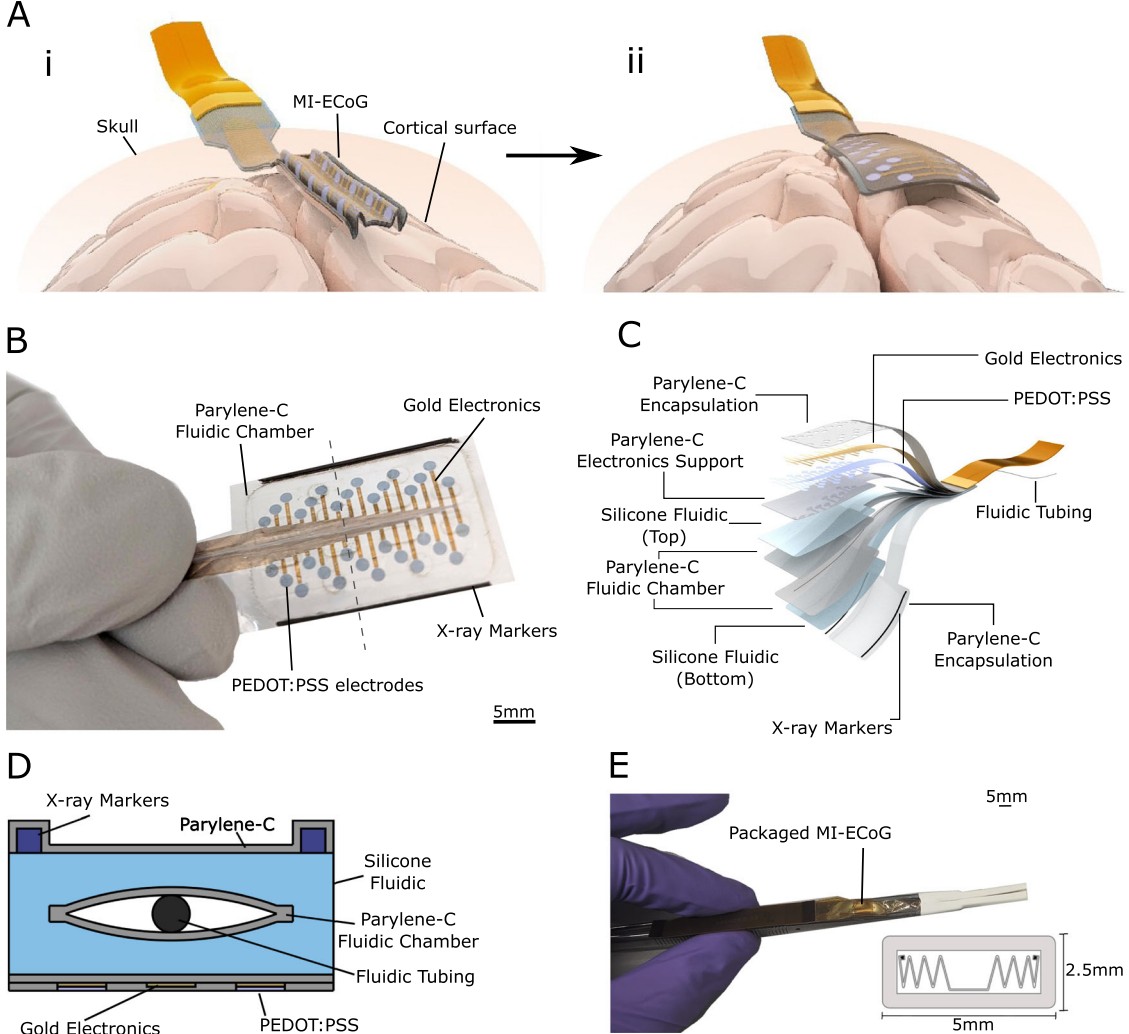

**Fig. 1 | Overview and design of the origami-inspired minimally invasive electrocorticography system. A** Illustration of the MI-ECoG concept: (**i**) A folded ECoG device is implanted onto the cortical surface using a burr-hole craniotomy (**ii**) Expanded to using fluidic actuation provide large-area cortical sensing. **B** An overview of the fabricated thin film MI-ECoG device. The dashed line indicates the location of the 2D cross-section shown in Fig. 1D. **C** An exploded cross-sectional view of the structure of the MI-ECoG device. **D** A 2D cross-sectional diagram of the MI-ECoG device. **E** Illustration of the custom implantation tool used to package and insert the MI-ECoG. A cross-sectional illustration (not to scale) from the front of the tool shows the illustration of a folded MI-ECoG device packaged within the custom introduction tool.

hydrogel of 6%/0.85% polyvinyl alcohol/phytagel was used which has been reported to have similar mechanical properties to porcine brain tissue[43] (Fig. 2B). A thin layer of 4% agarose was placed on top of the brain hydrogel to mimic the properties of the dura. The MI-ECoG was tested in the brain phantom by inserting and expanding the device below the 'dura' on top of the composite hydrogel layer. The depression on the 'brain' by the MI-ECoG was monitored in the model, with a maximum depression of 2.5 mm during insertion using the custom tooling.

By packaging the device with an origami-inspired concertina fold, fluidic actuation could drive the full expansion in the in-vitro model (Fig. 2C, D). To achieve full expansion in-vitro, a constant air pressure of 16–17 kPa was applied manually using a syringe (Fig. 2E), less than half of the maximum pressure these devices are capable of containing. Using the hydrogel phantom model, the expansion of wider 50 mm designs of the MI-ECoG could be tested, demonstrating the application of the platform for larger ECoG designs (Fig. 2F).

### Porcine in-vivo validation of the MI-ECoG
After successfully testing the expansion of the MI-ECoG in the in-vitro brain phantom model, the device was translated to porcine models.

Commercially bred pigs, around 50 kgs, were used to test the implantation and expansion of the MI-ECoG device on the surface of the cortex. For this, the skin on the surface of the skull was partially removed and two 12 mm burr-holes were drilled to allow implantation of two MI-ECoG devices—one for each brain hemisphere. After the craniotomy, a small, 6 mm incision was made in the dura, allowing the fully packaged MI-ECoG to be inserted onto the surface of the cortex using the custom insertion tool described above (Fig. 3A). The small incision in the dura helped to prevent unwanted swelling of the brain and reduced cerebrospinal fluid (CSF) leakage through the implantation site in comparison to a full durotomy[44]. The custom tooling allowed for repositioning the device whilst maintaining the tightly packaged concertina folding if an incorrect placement was observed under imaging or surgical feel of the tool with sliding on the cortex. Saline was briefly applied to help ensure the tool glided on the cortical surface without damage. To accurately test the expansion of the MI-ECoG, we ensured the device was implanted wholly underneath the skull using the custom insertion tool. This also helps ensure good cortical contact between the expanded MI-ECoG and the brain is achieved due to the intracranial pressure between the brain and the skull.

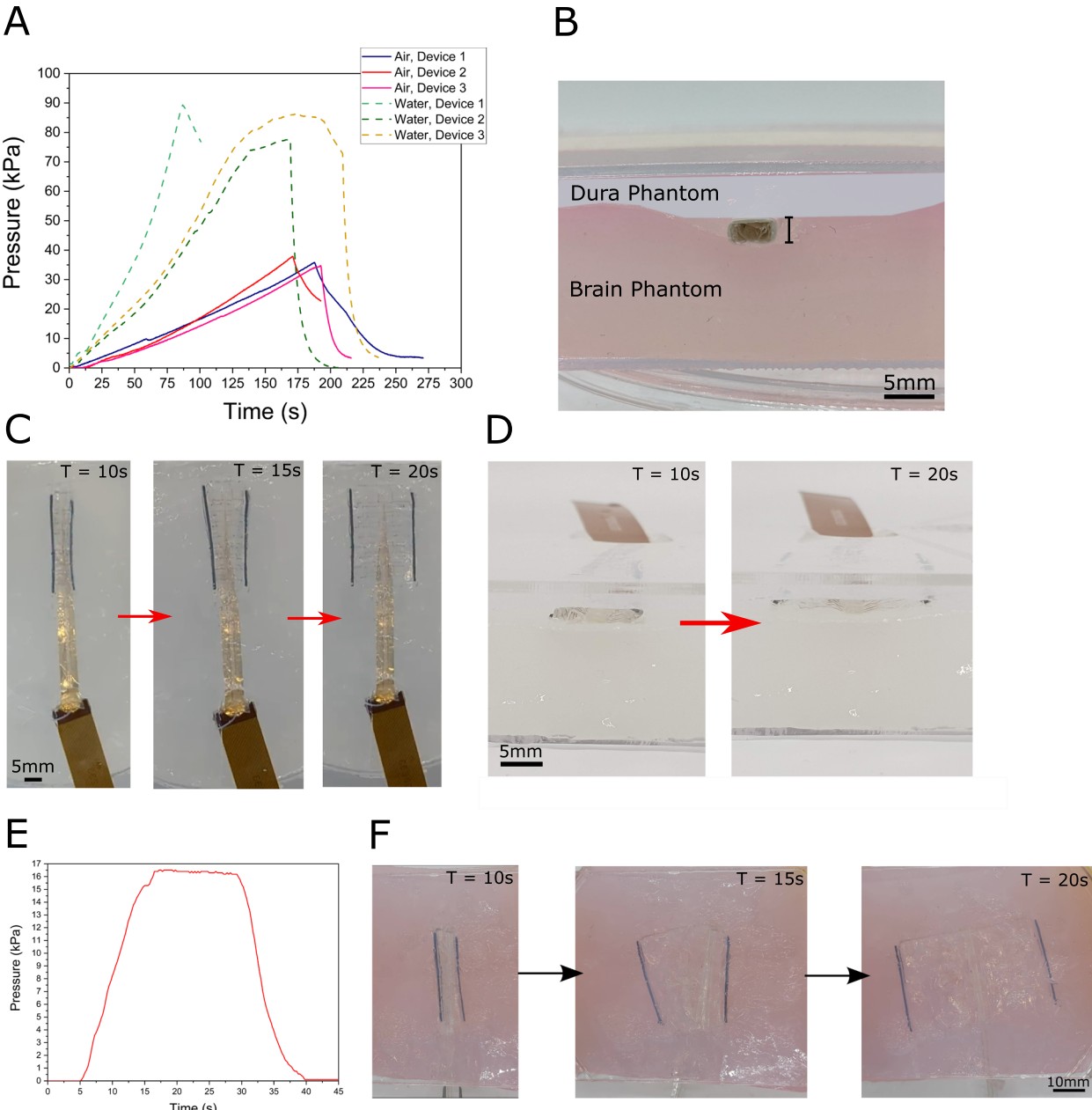

**Fig. 2 | In vitro characterisation of the MI-ECoG system in a hydrogel brain phantom. A** Time graph of the MI-ECoG devices being inflated using air or water at 1 ml/min until device failure. **B** Illustration of the in-vitro brain phantom used to test the expansion of the devices. The MI-ECoG is placed on the surface of the brain phantom within the custom implantation tool, demonstrating the maximum displacement of the phantom. **C** Top-down view of the MI-ECoG expanding in-vitro within the brain phantom model, using air manually applied using a syringe to provide fluidic pressure. **D** Side view of the device before and after expansion in the brain phantom. After expansion, the pressure is removed from the device which conforms the device to the hydrogel model. **E** Air pressure applied to the MI-ECoG during expansion in-vitro. During expansion, the pressure was manually held at 16–17 kPa, with the pressure removed after deployment to enable device conformity to surrounding tissue. **F** Time-series images showing the expansion of a 50 mm wide MI-ECoG in the brain phantom model using -16 kPa pneumatic pressure applied using a syringe.

Once the folded MI-ECoG was inserted onto the cortical surface and the insertion tool removed, a syringe was connected and coupled to a pressure transducer. The devices were manually inflated with air to drive expansion in the subdural space, and the device was X-ray imaged before and after deployment. Using air pressure, the device was fully expanded onto the cortical surface from the initial 4 mm width confined within the insertion tool to the full width of 20 mm (Fig. 3B), with the extent of expansion tracked through the X-ray fluoroscopy screening taken during expansion. Lateral projections were also taken to help confirm placement and ensure full expansion of the devices (Supplementary Fig. S3).

To achieve expansion, air pressure was manually applied at 14–17 kPa, similar to the pressure identified for expansion in-vitro, and significantly beneath the maximum air pressure tolerance identified previously (Fig. 3C). Air was chosen to actuate the device, both due to the compatibility with the pressure recording system and the ease of removal after expansion to flatten the device to increase conformability to the cortex. During the deployment procedure, the pressure was removed from the device mid-procedure to allow time to check the expansion distance using X-ray fluoroscopy. When the total expansion distance was observed to not be the full width of the device, more pressure was applied to ensure full expansion before

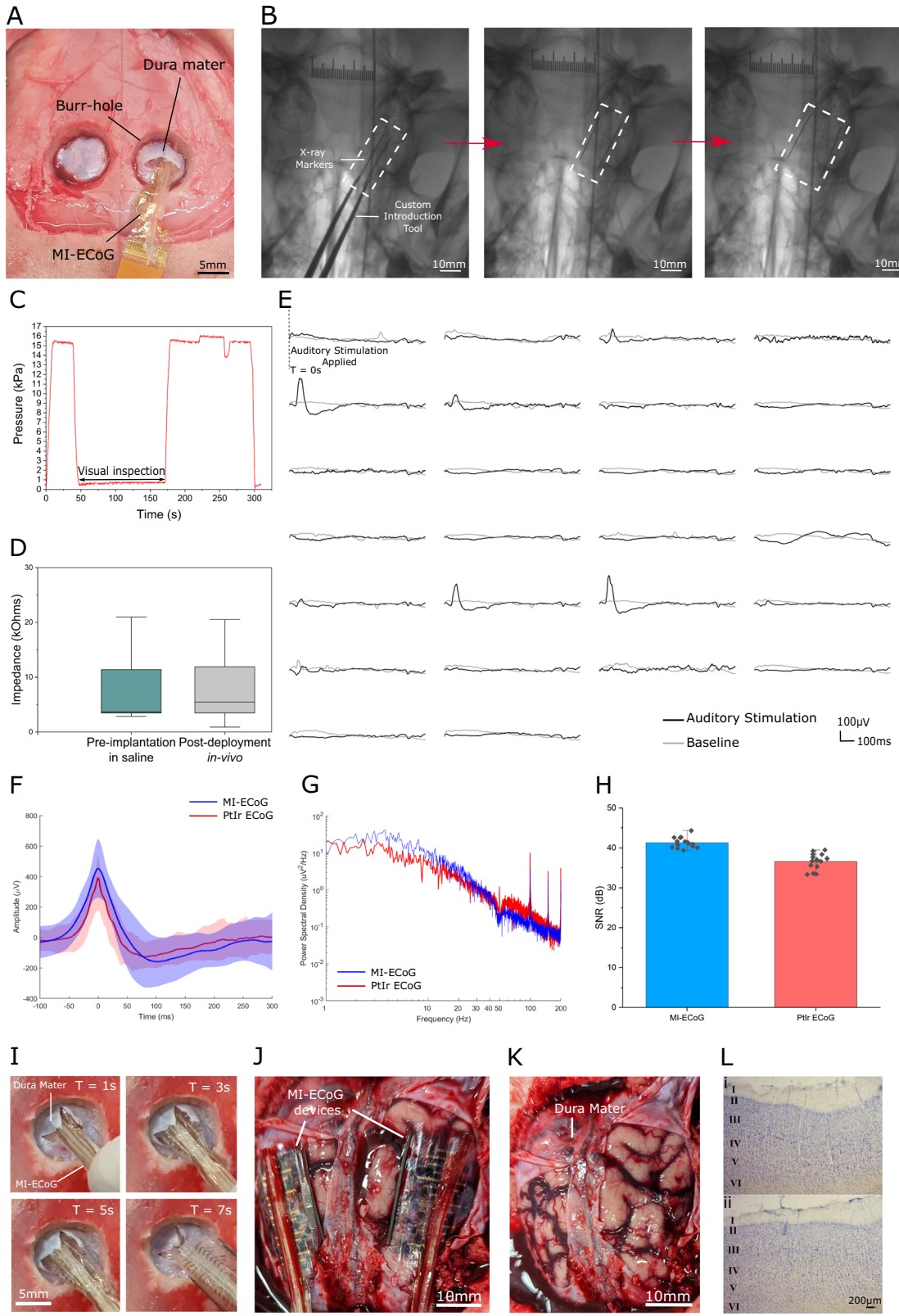

confirmation with X-ray imaging. After full expansion was confirmed, the air pressure was removed from the device to reduce the thickness of the device and help increase cortical conformity prior to electrophysiology recordings.

Following successful implantation, the fully deployed MI-ECoG was used to record cortical brain activity. The MI-ECoG was connected to an Intan RHS Stimulation/Recording system, and a reference gold electrode was placed subcutaneously above the skull. A 16-channel clinical grade PtIr ECoG was also surgically implanted onto the cortex of an anesthetised porcine using a traditional surgical craniotomy to provide a clinical comparison to the MI-ECoG (Supplementary Fig. S4). To quantify the impact of packaging and shape actuation on the MI-ECoG device, the impedance of the electrodes was measured at 1 kHz post-fabrication in saline and in-vivo

**Fig. 3 | In vivo implantation and expansion of an MI-ECoG device in an acute porcine model. A** Illustration of an MI-ECoG device being deployed subdurally onto the cortex in a porcine model. The device is being deployed through a burr-hole craniotomy, with a small slit cut in the dura to allow insertion. **B** X-ray images of the MI-ECoG device before, during and after expansion of the device, with the white boxes showing the outline of the device tracked using the X-ray opaque markers. At the beginning of the deployment, the custom introduction tool can be seen in the X-ray fluoroscopy containing the folded MI-ECoG. **C** Air pressure is manually applied to the device using a syringe to deploy the MI-ECoG. During expansion, the pressure was manually maintained at 14–17 kPa and was temporarily removed to allow visual inspection. **D** Impedance of the working electrodes ($n = 26$) measured at 1 kHz pre-implantation in saline and post-deployment on the cortex. Plot indicates median (middle line), 25th, 75th percentile (box), and 5th and 95th percentile (whiskers). **E** Recordings of induced AEPs (shown in black) from the surface of the cortex post-expansion from the deployed MI-ECoG electrode array, in comparison to baseline recordings (shown in grey). **F** Mean AEPs recorded by a representative channel on both the MI-ECoG (shown in blue) and a traditionally implanted PtIr ECoG (shown in red). Data are presented as mean values +/- SD. **G** Mean Power Spectral Density response of the AEP recordings obtained by the MI-ECoG (shown in blue) and the PtIr (shown in red). **H** Signal-to-noise ratio of the AEPs recorded by both the MI-ECoG (shown in blue) and PtIr ECoG (shown in red). Data are presented as mean values +/- SD ($n = 15$). **I** Time series of a deployed MI-ECoG device being explanted through the burr-hole craniotomy. This procedure was repeated in four animal experiments, including on the cortex used for histology. **K** Device placement of two MI-ECoG devices on the surface of the cortex after deployment with fluidic shape actuation. **J** Image of the brain after deployment with the dura and MI-ECoG devices removed. **L** Histology showing stained slices of the cortex from the MI-ECoG implantation site from the (**i**) Left Hemisphere and the (**ii**) Right Hemisphere of the cortex. The cortical layers are labelled accordingly.

post-deployment onto the cortical surface (Fig. 3D). Post-deployment, whilst the successfully fabricated electrodes are still conductive and in contact with the cortical surface, the mean impedance of the electrodes slightly increased from $7.87 \pm 7.61\,\text{k}\Omega$ to $9.17 \pm 9.14\,\text{k}\Omega$, within the accepted range for ECoG recordings[45]. To verify functionality, we recorded auditory evoked potentials (AEPs) from the cortex using the Intan RHS system to demonstrate neural recordings from the anesthetised porcine. The AEPs were induced using a loud, instantaneous auditory stimulus from across the surgical theatre. AEPs were present on several channels compared to baseline recordings, indicating these electrodes were covering the region containing the auditory cortex, with other channels not recording any evoked potentials (Fig. 3E). A similar spatial response was observed from the PtIr ECoG (Supplementary Fig. S4). The AEPs recorded by the MI-ECoG have similar profiles to previously described signals in anesthetised pigs[46], with a similar AEP response recorded by both the MI-ECoG post-expansion and the PtIr ECoG placed manually on the cortical surface (Fig. 3F). The averaged power spectrum density of baseline and AEP recordings from both the MI-ECoG and PtIr shows minimal difference between the recordings from the traditionally implanted PtIr grid and the MI-ECoG post-expansion (Fig. 3G). The MI-ECoG also exhibited a similar signal-to-noise ratio (SNR) of $41 \pm 1.8$ dB compared to the SNR of $36.9 \pm 2.5$ dB for the PtIr when recording AEPS after expansion (Fig. 3H). The slight increase in SNR in the MI-ECoG can be attributed to a slightly lower noise floor in the electrophysiological recordings by the device. A second device could be also implanted on the other hemisphere to enable the potential for full cortical mapping (Supplementary Fig. S4). Critically, after full deployment of the device, it was demonstrated the MI-ECoG could be explanted by sliding the device back through the slit made in the dura used for insertion (Fig. 3I). This was possible as the thin-film materials used in the design enable the device to self-compress to fit back through the insertion hole after full deployment.

MI-ECoG devices implanted in a separate pig were left in place ahead of a full craniotomy. This allowed us both to identify the final positioning of two MI-ECoG devices implanted on both sides of the cortex post-deployment and to evaluate the impact of the MI-ECoGs on the brain surface. When the skull was fully removed, the MI-ECoG was shown to be fully expanded under the dura, with the final position down the side of the right hemisphere (Fig. 3J). This corroborates the full expansion of the device after actuation inferred from the X-ray imaging. With the dura fully removed, the cortical surface was examined for any signs of damage caused by the implantation and expansion of the MI-ECoG (Fig. 3K). Visual inspection showed no visible damage was identified underneath the deployment site of the MI-ECoG. Cross-sectional imaging of the cortex beneath the deployment site after extraction showed no visible morphological changes or damage, with the thickness of the cortex in regions being unchanged or non-significantly reduced

(Fig. 3L). Cortical bleeding as a result of rupturing tissue during deployment was also not observed in the neuronal imaging.

## Discussion

In this work, we leverage soft-robotic technologies in clinically challenging environments for minimally invasive neural interfaces. When applying our shape-changing paradigm to the brain, a key challenge is device actuation in a mechanically challenging environment. We have shown that through improvements in the fluidic design, we are able to introduce an origami-inspired folded ECoG subdurally through a burr-hole, before enabling large-area cortical coverage through shape-actuation. Once deployed, our MI-ECoG fully expands to cover a cortical area of 600 mm$^2$. The dimensions of the MI-ECoG were limited only by the size and surgical access of the porcine models available, with larger MI-ECoG designs being able to be tested in hydrogel brain phantom models only. Our MI-ECoG was able to deploy a 32-electrode PEDOT:PSS/gold array onto the surface, where AEPs consistent with previous work[46] were recorded from the cortex, with no visible changes to the implantation site. The recordings of AEPs obtained by the MI-ECoG were comparable to recordings obtained using a traditionally implanted, 200 μm thick silicone PtIr foil ECoG, with similar power density spectrum profiles and SNR to the recordings of AEPs obtained with the PtIr device. This demonstrates that the MI-ECoG had good contact with the cortical surface post-expansion and is able to achieve similar electrophysiological performance to a traditionally implanted array. After deployment, the thin-film design of the MI-ECoG allows the removal of the device through the same entry burr-hole. This could prevent further surgical craniotomies from being performed on a patient after clinical monitoring.

Whilst we have demonstrated deployment of an MI-ECoG with gold-based thin-film electrodes for cortical recording, the advantage of our system is that the fluidic deployment platform is independent of the electrical design. The thin-film electrode materials and layout can be tailored to specific clinical applications, for example, electrodes constructed entirely out of PEDOT:PSS could be better suited for stimulation applications[47]. Testing the device with more standard ECoG fabrication approaches, such as the use of patterned metal foils, could further enhance the translatability of the expanding MI-ECoG concept. Further developments in the MI-ECoG design would need to ensure compatibility with further surgical imaging and guidance techniques that would be used during or after device implantation, such as X-ray fluoroscopy, CT, and MRI. Higher-density electrode layouts could also be explored, however, this would have to be complementary with innovations in high-density, thin film connections that would both not hinder device expansion and maintain the minimal dura opening. Compatibility with robotic and surgical neural navigation systems could also be explored.

The success of device expansion relied on the robustness of the soft robotic design and the device packaging for deployment, found

through in-vitro testing in a hydrogel model. Our previous work on shape-actuation for the spinal cord relied on the bonding of two layers of PDMS to provide support for fluidic actuation. By developing an actuation chamber made through bonding two layers of parylene-C, the maximum air pressure tolerance of the soft devices was increased to 30–35 kPa from the previously reported 8–12 kPa for the PDMS-based device. The increased pressure tolerance of these devices allows for their reliable expansion in confined environments that require pressures greater than was achievable with our previous devices. Due to parylene-C's increased resistance to deformation in comparison to PDMS, there was a significantly reduced fluidic failure rate of the devices during both fabrication and in-vitro testing by reducing the number of pin-hole defects and tears during fabrication. This optimisation has led to a fluidic fabrication yield of >95%. The overall thickness of the device, whilst thicker than some other thin-film ECoG designs, is thinner than clinical ECoG devices and other silicone ECoG devices[18,41]. This helps improve the cortical contact of the MI-ECoG post-expansion to enable high-quality electrophysiological recordings.

Packaging methodology was an important optimisation route investigated both in-vitro and in-vivo when developing these shape-changing devices. The requirement of expanding the devices in the virtual subdural space presented challenges in minimising the implantation footprint. By concertina folding the sides of the device, movement of the device was confined to a single plane, enabling expansion across the cortical surface. However, to enable clinical translation of the device, further refinement in the deployment tool is required to be suitable for use in a surgical setting, with the aim to improve usability and minimise deployment failures as a result of poor implantation procedure. A reduction in tool size could enable compatibility with new types of minimally invasive surgical techniques, such as a micro-slit craniotomy[24]. This will be the subject of future research alongside the integration of improved electrode connections that allow compatibility with surgical anchoring to the surrounding tissue when used for chronic recordings.

We have shown that with soft robotic design and selective packaging techniques, large-area shape-changing neural interfaces can be deployed in mechanically challenging environments. In comparison to previous shape-actuating neural implants, our approach mimics the surgical implantation approach of ECoG strip devices[48,49], with the aim to further reduce the implantation footprint required to become compatible with a micro-slit implantation approach. We have demonstrated our origami-inspired packaging approach enables the MI-ECoG to have a 5× increase in cortical area coverage after actuation in comparison with the coverage achieved during initial implantation. The demonstrated cortical coverage here is 6.7× larger than the eversion soft robotic ECoG strip described by ref. 41 whilst only using a 6 mm slit in the dura for insertion. During the testing of our device, the skull was left fully in place, with no pre-treatment to affect brain volume to ensure fully representative surgical conditions with our porcine model. Despite this, we were still able to demonstrate a 2× larger cortical coverage to another unfolding ECoG design that required a full craniotomy[44]. The unfolding approach to expansion, as it is independent of device width, enables the geometry of the device to be tailored to the clinical footprint required for either short-term large-area clinical mapping, or longer-term recording from coverage of specific cortical regions. In the future, we envision that the MI-ECoG implantation and deployment strategy will enable device designs that integrate electrode layouts already applied in clinical use as well as novel microelectrode designs. Our robust soft-robotic actuation chamber design could also be tested with alternative origami-inspired folded structures with the aim of further minimising the initial implantation footprint.

Through this work, we show that micron-thin biocompatible materials combined with rapid prototyping and soft-lithography techniques can make existing neural interface architectures suitable for minimally invasive implantation. Through careful design of the implant geometry, novel bioelectronic devices can be engineered to reliably expand in complement to the surrounding tissue thereby reducing surgical side effects and complications. We anticipate this paradigm in minimally-invasive design will enable wider clinical use of ECoG as well as new routes of bioelectronic medicine and new applications of ECoG previously unexplored due to surgical risk.

## Methods
### Fabrication of MI-ECoG
To form the fluidic chamber in the centre of the device, a 4 μm layer of parylene-C was initially deposited onto a glass slide coated in 3% soap solution (Micro 90, VWR). Another layer of 3% soap solution was spin-coated on the parylene-C at 1000 rpm for 30 s before a second 4 μm layer of parylene-C was deposited. The chamber was then laser cut from the two layers of parylene using a 30 W, 10.6 μm $CO_2$ laser cutter (VLS2.3, Universal Laser Systems). To form the body of the fluidic, PDMS (Sylgard 184, Dow) was mixed at a 1:10 ratio of curing agent to silicone base, before being spin-coated onto a glass slide coated with 4 μm parylene-C at 3000 rpm for 90 s. The silicone was partially cured at 60 °C for 3 min before the laser-cut fluidic chamber was placed on top. This was capped with further PDMS by blade coating using a 50 μm spacer. X-ray opaque markers were formed by mixing Bismuth powder (mesh 100, Sigma-Aldrich) with PDMS at a Bi:PDMS 1:4 w/w ratio and cured in a 3D printed mould. These were placed on the sides of the device before a full cure at 90 °C for 1 h. The device was treated with a silane solution (A-174, Sigma-Aldrich) before a 4 μm layer of parylene-C was deposited.

After releasing from the substrate, an incision was made in the neck of the fluidic chamber, and a Polytetrafluoroethylene (PTFE) fluidic tube (Bohlender 0.4 mm outer diameter/0.2 mm inner diameter, VWR) was inserted into the centre of the device. This tube was capped and small holes pierced down the length before insertion. The device was then sealed by injecting UV curing adhesive (Loctite 358, RS).

The electronics were fabricated using standard photolithographic processes. Briefly, AZnlof 2035 (Merck) photoresist was spin-coated onto a Silicon wafer coated in 2 μm parylene-C at 3000 rpm for 30 s before a soft bake for 60 s at 100 °C. The wafer was exposed to ultraviolet (UV) light (8 s, 80 mJ/cm$^2$) using a MA/BA6 contact mask aligner (Karl Suss) through a plastic photomask (JD Photodata). A second baking step was applied for 60 s at 115 °C before development in AZ726 (Merck) for 120 s. After development, the samples were rinsed using deionised (DI) water and dried using compressed nitrogen.

The substrate with developed resist was activated with $O_2$ plasma (Diener Pico) at 0.6 mbar for 60 s at 100 W. 10 μm Ti and 100 μm Au were deposited sequentially on the activated substrate under vacuum ($<6 \times 10^{-7}$ mtorr) in a PVD75 E-Beam system (Kurt Lesker), with a 10 min rest period between each deposition. The substrates were soaked in acetone for 1 h to perform lift-off, before spraying with acetone and gentle wiping with a sponge to remove excess resist. The substrates were washed with isopropyl alcohol before inspection to ensure complete lift-off under a microscope.

A second 2 μm layer of parylene-C was deposited on the substrate before AZ 10xt (Merck) photoresist was spin-coated at 3000 rpm for 30 s. After a soft bake at 110 °C for 120 s, the substrate was exposed to UV light (20 s, 80 mJ/cm$^2$) and then developed in 1:4 AZ400k:DI water solution for 4 min. The outline of the device was then etched using Reactive Ion Etching (Oxford Plasma Pro 80 RIE, 160 W, 50 sccm O2, 5 sccm CF4) before the remaining resist was washed off with acetone. A 3% soap solution (Micro 90, VWR) was spin-coated onto the wafers at 1000 rpm for 30 s before a 2 μm sacrificial layer of parylene-C was deposited. AZ 10xt photoresist was developed using AZ 400k once

again used to pattern the electrodes and contact pads, with the metal exposed with Reactive Ion Etching.

A prepared PEDOT:PSS solution was spin-coated once at 1500 rpm for 30 s and twice at 3000 rpm for 30 s with a 60 s soft-bake at 110 °C between spin-coatings. To prepare the PEDOT:PSS solution, commercially available PEDOT:PSS solution (Heraeus Clevios PH1000) was mixed with 5% v/v ethylene glycol (Sigma-Aldrich), 0.05% wt. dodecyl benzene sulfonic acid (Sigma-Aldrich), and 1% 3-Glycidyloxypropyl trimethoxysilane (GOPS) (Sigma-Aldrich). The solution was filtered using a PTFE 0.45 μm membrane filter. After spin-coating, the sacrificial layer of parylene-C was peeled off, leaving PEDOT:PSS on the electrode surface. The substrate was baked for 1 h at 120 °C to fully cross-link the PEDOT:PSS.

The electronics were bonded to an FFC cable (RS) using a Finetech bonder (Pico Ma) and ACF tape (Hitachi). The electronics were finally bonded to the fluidic device using 5 μm double-sided adhesive. The sides of the device were folded using 1 mm card spacers and packaged within the custom insertion tool. The tool was made from 120 mm tweezers with 1 mm thick PVC heat shrink (RS) moulded around a 3D printed guide, before being affixed to the tweezers using adhesive. During packaging, a 2 × 1 mm PDMS spacer was placed down the centre of the device to aid in deployment.

## Pressure testing
To measure the fluidic pressure whilst testing, the MI-ECoG devices were connected to a syringe via a 3D-printed t-connector that was connected to an MXP5100GP pressure sensor (NXP), with the pressure data recorded via an Arduino Mega (Fig. S5). To characterise the maximum pressure of the devices, a 10 ml syringe pump (KD Scientific) was connected to the system and the devices were inflated using either air or DI water at 1 ml/min. The experiments were stopped after a loss of pressure indicated by a pressure drop.

## In-vitro brain phantom testing
The hydrogel brain phantom was made of a 1:1 w/w ratio of two hydrogels, polyvinyl alcohol (PVA) and Phytagel. 6% w/w of 146,000–186,000 molecular weight PVA (Sigma-Aldrich) was mixed into DI water and stirred at 90 °C for 1 h until fully dissolved. 0.85% w/w Phytagel (Sigma-Aldrich) was mixed into DI water and stirred at 90 °C for 1 h until fully dissolved. The two hydrogels were then mixed at 90 °C for 30 min. The solution was poured into a 100 mm petri dish and frozen for 24 h. After thawing, a thin layer of 4% w/w of Agarose (Sigma-Aldrich) dissolved in DI water was poured onto the surface until level with the top of the petri dish. The petri dish was cut to enable device implantation and secured to the base to create the brain phantom.

After the device was implanted between the hydrogel layers in the brain phantom, the device was connected to a syringe through the pressure testing setup and manually inflated until fully expanded.

## Porcine model testing
Six implantations of the MI-ECoG were performed, two separate implantations in porcine cadavers directly post-euthanasia, and four devices in acute in-vivo porcine models.

The in-vivo testing was carried out at the porcine experimental facility of the Department of Veterinary Medical Sciences of the University of Bologna (52-2004-A, since 27/05/2004), and at the Bancroft Centre, University of Cambridge.

All the procedures at the University of Bologna were performed in accordance with the current European Legislation and were approved by the Italian Ministry of Health, as dictated by the legislative decree 26/2014 (approval number 434/2023-PR). At the University of Cambridge, the animal procedures were carried out in accordance with the UK Animals (Scientific Procedures) Act, 1986. Work was approved by

the Animal Welfare and Ethical Review Body of the University of Cambridge, and was approved by the UK Home Office (project licence no. PP6076787).

Six commercial hybrid pigs (Large White × Landrance × Duroc), with a mean body weight of 50 kg, were enrolled in the study. Prior to the procedures, animals were considered healthy on the basis of clinical examination. Sex was not considered in this study.

For the two implantations of the MI-ECoG in porcine cadavers, the animals were euthanized for tissue samplings as sham controls for other experimental protocols approved by the Italian Ministry of Health, as dictated by the legislative decree 26/2014. Briefly, pigs (body weight 30–45 kgs) were deeply sedated intramuscularly with a mixture of tiletamine-zolazepam (3 mg/kg) and dexmedetomidine (0.015 mg/kg) and then euthanized upon intravenous barbiturates overdose (thiopental sodium, 60 mg/kg). For the experimental preparation, immediately after euthanasia, cadavers were placed in sternal recumbency, and the skin covering the frontal bone was removed. Burr holes were drilled using a craniotome equipped with 3 mm carbon steel ball drill bits, 5 mm above a line connecting the dorsal margins of the orbits and 5 mm laterally to the sagittal midline. Once exposed, the dura mater was gently lifted using anatomical forceps and cut with a #11 scalpel blade. Device insertion was performed under fluoroscopy guidance (Arcadis Avantic, SIEMENS). The device was then connected via the pressure testing setup to a syringe and the device was manually inflated.

For the four implantations of MI-ECoG devices in acute porcine models, the pigs were deeply sedated intramuscularly with a mixture of tiletamine-zolazepam (3 mg/kg) and dexmedetomidine (0.015 mg/kg). Once lateral recumbency was achieved, an intravenous catheter (20 Gauge) was placed in the auricular vein, and general anaesthesia was induced with propofol (3–5 mg/kg). Pigs were then orotracheally intubated and connected to a mechanical ventilator to grant respiratory support and inhalational anaesthesia maintenance (Sevoflurane 2–4% in a 1:1 air/oxygen mixture). The respiratory rate was set to maintain normocapnia throughout the entire procedure. Systemic analgesia was achieved upon constant rate infusion of fentanyl (0.01–0.03 mg/kg/h), while fluid therapy (lactated ringer) was set at 6–8 ml/kg/h. Standard anaesthetic monitoring was set up, including SpO2, heart (HR) and respiratory (RR) rates, capnometry (CO2) and capnography, non-invasive blood pressure, and rectal temperature (T). For the entire procedure, animals were placed in sternal recumbency. Upon skull exposure, burr holes were drilled using a neurosurgical power system (ELAN 4, Aesculap) equipped with a 12/15 mm Hudson craniotome burr, 5 mm above a line connecting the dorsal margins of the orbits and 5 mm laterally to the sagittal midline. Once exposed, the dura mater was gently lifted using anatomical forceps and cut with a #11 scalpel blade. Device insertion was performed under fluoroscopy guidance (Arcadis Avantic, SIEMENS). The device was then connected via the pressure testing setup to a syringe and the device was manually inflated.

To provide a clinical comparison a silicone, 16-channel PtIr ECoG (Cortec GMBH) was introduced manually onto the cortex. A craniotomy was performed above the Right Hand Hemisphere of the cortex, with the dura mater opened. The device was then manually placed onto the surface of the cortex, with good cortical contract ensured. The ECoG was connected to a 32-channel RHS stim/record head stage (Intan) for electrical characterisation and electrophysiology recording.

To avoid pharmacological interference during electrophysiology recording, fentanyl infusion was stopped 10 min before, and anaesthesia maintenance was switched from inhalational to TIVA (total intravenous anaesthesia) achieved by means of propofol (0.1–0.2 mg/kg/min). Respiratory support was maintained.

At the end of the procedure, animals were euthanized by means of barbiturate overdose (thiopental sodium, 60 mg/kg). Upon death

confirmation, a full craniotomy was immediately performed to expose and collect the brain, which was fixed in formalin.

## Electrical characterisation

To electrically characterise the device, an Intan 128-channel RHS Stimulation/Recording system was used with IntanRHX v3.0.4. The MI-ECoG was connected to the system with a 32-channel RHS stim/record head stage (Intan) with a custom PCB. Impedance was measured at 1 kHz, with the device pre-implantation measured in phosphate-buffered saline (PBS) solution (0.01 M, Sigma-Aldrich) with a Gold reference electrode. For the post-deployment impedance measurement, a subcutaneous pocket was created underneath the skin of the porcine model, with a Polyethylene terephthalate (PET) film covered with Au inserted underneath as a reference electrode. Electrophysiology recordings were acquired in the same setup. Electrophysiology recordings were carried out at a sampling rate of 30 kHz, followed by filtering using a notch filter at 50 Hz and a 0.1–200 Hz bandpass filter. The AEPs were stimulated by a loud clap created from across the surgical theatre. The electrical setup used in the porcine testing is shown in Fig. S5. Electrochemical data was acquired using a Palmsens 4 with PSTrace 5.9. Data analysis and plotting were performed with Matlab2021a.

## Histology

Smaller blocks, containing the involved cortex, were cut from the formalin-fixed brains, washed in PBS (pH 7.4), cryoprotected in 20% glycerol in 0.02 M PBS (pH 7.4) at +4 °C for 48 h, frozen in dry ice, and stored at −70 °C. 15 μm thick frozen coronal sections throughout the entire rostrocaudal extent of the involved cortex were cut with a sliding microtome. The sections were stained with thionin as follows. Sections were taken out of the 10% formaldehyde solution, mounted on gelatin-coated slides, and dried overnight at 37 °C. Sections were defatted 1 h in a mixture of chloroform/ethanol 100% (1:1), and then rehydrated through a graded series of ethanol, 2 × 2 min in 100% ethanol, 2 min in 96% ethanol, 2 min in 70% ethanol, 2 min in50% ethanol, 2 min in dH2O, and stained 30 s in a 0.125% thionin (Fisher Scientific) solution, dehydrated and coverslipped with Entellan (Merck, Darmstadt, Germany). of sections to be stained with thionin was stored in 10% formalin. The thickness of the cortex and of each cortical layer was measured on thionin-stained coronal. Images were captured using a digital camera (AxioCam ERc5s®, Zeiss, Germany), and linear measurements were performed using the AxioVision Rel.4.8 software (Zeiss). For each layer and for the entire cortex, 3–6 measurements were taken in each cortical area.

## Reporting summary

Further information on research design is available in the Nature Portfolio Reporting Summary linked to this article.

# Data availability

All data supporting the findings of this study are available within the article and its supplementary files. Any additional requests for information can be directed to, and will be fulfilled by, the corresponding authors. Source data are provided with this paper.

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

## Acknowledgements

L.C. acknowledges funding from the U.K. Engineering and Physical Sciences Research Council Centre for Doctoral Training in Sensor Technologies for a Healthy and Sustainable Future (EP/S023046/1). M.L.B. and D.V. acknowledge funding from the University of Bologna (RFO programme 2021–2022). A.C.L. acknowledges funding from the University of Cambridge Borysiewicz Fellowship program. B.J.W. acknowledges funding from the Engineering and Physical Sciences Research Council Centre for Doctoral Training in Sensor Technologies and Applications (EP/L015889/1). J.G.T. is supported by the National Institute for Health Research Invention for Innovation award (NIHR203355). M.M. is supported by the Biotechnology and Biological Sciences Research Council (BB/T009314/1). D.G.B. is supported by Health Education England and the National Institute for Health Research HEE/NIHR ICA Program Clinical Lectureship (CL-2019-14-004). S.E.H. acknowledge funding from the National Institute of Health Research (NIHR) (G112655) and is the Laureate of the Helaers Research Prize for Neurosurgery. C.M.P. acknowledges funding from the U.K. Engineering and Physical Sciences Research Council IAA award, and the Biotechnology and Biological Sciences Research Council David Phillips Fellowship (BB/T009314/1). The devices were built in the laboratory for prototyping soft neuroprosthetic technologies, funded by the Sir Jules Thorn charitable trust (233838).

## Author contributions

The project was initially conceived and initiated by L.C., C.M.P., B.J.W., G.G.M., and D.G.B. Device fabrication, development, and testing were carried out by L.C. In-vitro experiments were designed and analysed by L.C. Porcine experiments were carried out by L.C., C.M.P., A.C.L., M.L.B., D.V and A.E. Porcine surgery was carried out by S.E.H., D.V., and A.E. X-ray fluoroscopy imaging was performed D.V and A.E. A.C.L. and L.C. performed and analysed the electrical stimulation experiments and data. J.G.T. and M.M. provided the drawing renders of the device. L.C. produced the first draft of the manuscript, C.M.P., J.G.T., and L.C. edited the manuscript, and C.M.P. produced the final draft of the manuscript. C.M.P., D.G.B., and G.G.M. oversaw the project.

## Competing interests

C.M.P., D.G.B., G.G.M., and B.J.W. are inventors on a patent related to this work filed by Cambridge Enterprise Ltd (no. PCT/GB2020/051684, filed 13 July 2020). The remaining authors declare no conflict of interest.
