## [Peer Review File · Nature Communications]

REVIEWER COMMENTS

Reviewer #1 (Remarks to the Author):

In this paper, the authors demonstrated subdural large-area ECoG implant (MI-ECoG), enabling shape actuation through the integration of a fluidic chamber in the center of the implant. To achieve the fluidic platform of the MI-ECoG, the authors developed a new device stack-up and polymer laser welding technique, resulting in a device capable of sustaining the pressures required for shape actuation underneath the skull. In addition, to identify a suitable packaging methodology, and the fluidic pressures required for reliable expansion, the expansion of the MI-ECoG device was tested in-vitro using a hydrogel brain/dura model. Following that, in-vivo validation of the MI-ECoG was performed to ensure that it was functional.

However, I think there are several critical issues with this paper as follows.

(1) Although the concept of this work is interesting in that the skull was partially removed to insert the MI-ECoG into the surface of the brain and pressure was used to stretch the folded MI-ECoG to detect a large area of brain signal, the width of the stretched MI-ECoG is 10 mm and the custom implantation tool used to insert it is 5 mm, which does not dramatically increase the width.

(2) Fluidic tubing is inserted to use pressure, which makes the device 60-70 μm thick, which is thicker than other ECoG devices and can reduce the quality of detecting neuron signals.

(3) The auditory evoked potentials recordings and imaging data have not been quantitatively analyzed. As a result, assigning meaning to them is difficult.

Therefore, I do not think that this paper is of high quality enough to pass the high standard of Nature Communications.

Reviewer #2 (Remarks to the Author):

In this manuscript, Coles et al demonstrate an ECoG array capable of unfolding using microfluidics on the surface of the brain. The reasoning behind this operation is to reduce the cranial window size while maintaining a large spatial of coverage.

While the idea of a mechanically actuating neural interface device is interesting, there are several fundamental and technical issues in this manuscript.

- In the folded state the device is ~ 5 mm and it expands to a few cm after unfolding. The folded state of the device is too large to be used in a surgical burr hole that is typically done for SEEG electrodes (< 1.5

mm). Therefore, a cranial window is needed, and the implantation cannot be considered as a minimally invasive procedure. Next, the ratio of fold/unfolded is not sufficient. An ECoG array typically should cover multiple cortical areas to allow cortical mapping. This is more evident in the surgical picture (Figure 3F). There is a very small difference between the diameter of the skull opening vs. the probe. At this aspect ratio, most surgeons are able to fold a conventional ECoG array during insertion into the cranial window and allow it to open on the surface of the brain due to the elastic nature of the silicone substrate. I think that the authors should demonstrate the performance of their device using SEEG burr holes and demonstrate spatial coverage comparable to a clinical ECoG in order to state that their device presents a substantial advance over currently available technology.

- SEEG probes require anchor-bolts through which the electrodes are threaded. This is why the electrode are kept at a same or smaller diameter compared to their cables. The authors should describe the mechanism by which they aim to insert and secure the significantly larger and planar structure. No data has been shown on the stability of the device. Such data is important even for short acute recordings.

- A critical technical issue with intracranial devices is the reliable mechanical contact between the electrode and the surface of the brain. In small cranial windows there is often a significant flow of cerebrospinal fluid. What is the mechanism that maintains appropriate contact of the probe with the surface of the brain despite ongoing fluid flow? How do authors ensure the device is not detached and floating above the brain surface due to accumulating cerebrospinal fluid?

- The manuscript is very light on actual electrophysiological data. Figure 3D is the only sample traces that can be used to evaluate the spatial resolution of the device. The recordings do not have a consistent gradient of the auditory response amplitude; instead, there is a sparse representation of the evoked response. This could be due to spatially inconsistent and poor contact between electrodes and the brain as mentioned above. It also seems that only two trials have been performed, and only one resulted in a response. In order for the authors to support their claims they should at least compare their device with the performance of a traditional ECoG array.

Overall, this is an interesting concept and efforts in this direction are worthwhile, but the current manuscript requires extensive modification with additional experimentation and analysis before the authors' claims can be supported.

We thank the reviewers for their thorough review and offer the changes to the manuscript outlined below.

Reviewer 1

In this paper, the authors demonstrated subdural large-area ECoG implant (MI-ECoG), enabling shape actuation through the integration of a fluidic chamber in the center of the implant. To achieve the fluidic platform of the MI-ECoG, the authors developed a new device stack-up and polymer laser welding technique, resulting in a device capable of sustaining the pressures required for shape actuation underneath the skull. In addition, to identify a suitable packaging methodology, and the fluidic pressures required for reliable expansion, the expansion of the MI-ECoG device was tested in-vitro using a hydrogel brain/dura model. Following that, in-vivo validation of the MI-ECoG was performed to ensure that it was functional.

However, I think there are several critical issues with this paper as follows.

(1) Although the concept of this work is interesting in that the skull was partially removed to insert the MI-ECoG into the surface of the brain and pressure was used to stretch the folded MI-ECoG to detect a large area of brain signal, the width of the stretched MI-ECoG is 10 mm and the custom implantation tool used to insert it is 5 mm, which does not dramatically increase the width.

We thank the reviewer for their comments however it seems our findings were misunderstood. In this work, we demonstrate an expansion of the MI-ECoG from 4 mm width in the introduction tool to a 20 mm width. The fully expanded 20 mm x 30 mm size of this implant is the largest possible sub cranial coverage in a porcine model due to the anatomy of their skull and brain. Unfortunately, we are not able to access any animal models that would allow for demonstration of wider coverage. However, the underlying fluidic-actuation and origami packaging concept does allow for unfolding wider devices up to and beyond standard human clinical grid dimensions. To this end, we have included a demonstration of a 50mm wide MI-ECoG system expanding in our hydrogel brain/skull phantom model.

For clarification, we have included the following statement into the manuscript and added the 50 mm device demonstration to the Figure 2:

“Using the hydrogel phantom model, expansion of wider 50mm designs of the MI-ECoG could be tested, demonstrating the application of the platform for larger ECoG designs (Fig 2F).” (pg 5, paragraph 3)

(2) Fluidic tubing is inserted to use pressure, which makes the device 60-70 μm thick, which is thicker than other ECoG devices and can reduce the quality of detecting neuron signals.

Whilst we acknowledge that our device is thicker than some thin-film devices reported in the literature, our device is still thinner than other silicone based ECoGs and is notably thinner than clinically approved ECoG arrays (see reference 18 and 36 in the revised manuscript).

To better understand if the device thickness is a limitation, we performed additional experiments in pigs to allow for direct comparison of electrophysiology recordings from our MI-ECoG relative and a state-of-the-art commercial clinical ECoG. Specifically, the quality of recordings from our MI-ECoGs implanted via burr hole were compared to a 200 μm thick PtIr ECoG “AirRay Electrode” device from Cortec GMBH implanted directly on the brain surface following a full craniotomy. The AirRay Electrode is, to our knowledge, the most compliant ECoG electrode that is FDA approved and

commercially available. As shown in the revised Figure 3, our MI-ECoG electrode exhibits similar electrophysiological performance to the commercial device with a slightly better signal to noise ratio (SNR). We therefore contend that the MI-ECoG is sufficiently thin/flexible to allow for high quality electrophysiology recordings that can inform clinical practice.

To provide further context for the reader, we have included the following statement into the manuscript (Pg 11, paragraph 2).

“The overall thickness of the device, whilst thicker than novel thin-film ECoG designs, is thinner than clinical ECoG devices and other silicone ECoG devices. This helps improve the cortical contact of the MI-ECoG post-expansion to enable high quality electrophysiological recordings.”

(3) The auditory evoked potentials recordings and imaging data have not been quantitatively analyzed. As a result, assigning meaning to them is difficult. Therefore, I do not think that this paper is of high quality enough to pass the high standard of Nature Communications.

We thank the reviewer for the suggestion on how to improve our manuscript. As described above, we have performed additional in vivo experiments and analysis on the electrophysiology data including a comparison to a 200 μ m thick silicone Ptlr ECoG provided by Cortec GMBH, which is a state-of-the-art clinical grade device. We now show, through additions to Figure 3, that the performance of the MI-ECoG post-expansion is comparable to the traditional ECoG device. Specifically, the representative recordings of auditory evoked potentials (AEPs) are shown to have a similar profile, with the recordings obtained by both devices having a similar power spectral density between 0.1-200Hz. The recordings of the AEPs from both devices had a similar SNR, with a slightly higher SNR obtained by the MI-ECoG due to a lower noise floor. Further comparison between the devices is also illustrated in a revised Figure S4, where the AEP spatial response recorded using the Ptlr is shown. We hope the reviewer will agree that with these additions, the manuscript now lives up to the high standards of this journal.

Reviewer 2

In this manuscript, Coles et al demonstrate an ECoG array capable of unfolding using microfluidics on the surface of the brain. The reasoning behind this operation is to reduce the cranial window size while maintaining a large spatial of coverage.

While the idea of a mechanically actuating neural interface device is interesting, there are several fundamental and technical issues in this manuscript.

- In the folded state the device is \sim 5 mm and it expands to a few cm after unfolding. The folded state of the device is too large to be used in a surgical burr hole that is typically done for SEEG electrodes (<1.5 mm). Therefore, a cranial window is needed, and the implantation cannot be considered as a minimally invasive procedure. Next, the ratio of fold/unfolded is not sufficient. An ECoG array typically should cover multiple cortical areas to allow cortical mapping. This is more evident in the surgical picture (Figure 3F). There is a very small difference between the diameter of the skull opening vs. the probe. At this aspect ratio, most surgeons are able to fold a conventional ECoG array during insertion into the cranial window and allow it to open on the surface of the brain due to the elastic nature of the silicone substrate. I think that the authors should demonstrate the performance of their device using SEEG burr holes and demonstrate spatial coverage comparable to a clinical ECoG

in order to state that their device presents a substantial advance over currently available technology.

We thank the reviewer for their comments. In this work, we demonstrate an expansion of the MI-ECoG from 4 mm width in the introduction tool to a 20 mm width. This five times expansion is significantly greater than what is possible with any existing commercial ECoG electrodes and as such would represent a significant advancement in clinical care. Key to the success of the MI-ECoG was the development of controlled, origami inspired packaging to minimise the initial footprint. Other methods to package the MI-ECoG were tested during the development phase, however concertina folding proved to be the only reliable approach for large, subcranial expansion.

Figure 3F in the original manuscript (Figure 3I in the revised manuscript) shows the explantation process during which the device is forced to compress laterally to fit through the dura incision and therefore should not be considered as a representation of the expanded state. We have added discussion in the manuscript that highlights how the expansion ratio is a significant advancement over recent high profile reports of other ECoGs that aim to reduce the invasiveness of implantation. Notably, the 6mm slit in the dura used for implantation is consistent with these recent reports while on the other hand our device provides a significantly larger cortical coverage post-expansion.

As we mention in response to Reviewer 1, it is also important to understand the limits of what can be done in large animal models. The fully expanded 20 mm x 30 mm size of the MI-ECoG implant tested in vivo is the largest possible subcranial coverage in a porcine model. Subcranial expansion of a larger device is not possible due to the anatomy of the porcine skull and brain. Reducing the size of the burr hole is also challenging due to the high thickness of the pig skull as well as the presence of their large sinus cavities. Unfortunately, we are not able to access any animal models that would allow for demonstration of wider coverage nor are we aware of any research animal models that would be more suitable for this stage of development than the pigs used in this study.

We have conducted trial implantations in frozen human cadavers however, we found the deteriorated mechanical properties of the atrophied cadaver brain and lack of CSF makes for a poor model of the subdural space in a living subject. While our human cadaver work did indicate it will be possible to reduce the size of the burr hole, given the limitations of the cadaver model, we do not feel it is appropriate to include those findings in this manuscript. We instead leave reduction of the burr hole size and packaging footprint of the implant for future work and have added discussion to that point in the revised manuscript.

We have added to Figure 2 the demonstration of a 50 mm wide device expanding in a brain/dura phantom model to highlight the capability of deploying larger MI-ECoG devices with the same device concept. In the future, we anticipate successful clinical deployment of the 20 x 30 mm MI-ECoG will allow for clinical trialling of larger arrays. In the mean time, we contend that reducing the surgical footprint for implantation of a 20 x 30 mm grid from $> 600 \text{ mm}^2$ to $< 20 \text{ mm}^2$ is already a major advancement over currently available technology that is likely to significantly improve the patient experience and tolerance of ECoG procedures. We also observe that one could implant multiple 20 x 30 mm arrays through the same entry point to allow for wider coverage.

To ensure an accurate representation of the MI-ECoG expansion, even if the implantation size is reduced, we ensured the device was implanted fully underneath the skull during testing.

For clarification the following statements have been added:

“To accurately test the expansion of the MI-ECoG, we ensured the device was implanted wholly underneath the skull using custom tooling.” Pg 7, paragraph 1

“During the testing of our device, the skull was left fully in-place with no pre-treatment to affect brain volume to ensure fully representative surgical conditions with our porcine model. Despite this, we were still able to demonstrate a 2x larger cortical coverage with a similar implantation profile to another unfolding ECoG design.” Pg 10, paragraph 2

“Our robust soft-robotic actuation chamber design could also be tested with alternative origami-inspired folded structures with the aim to further minimise the initial implantation footprint.” Pg 11 paragraph 4

“Using the hydrogel phantom model, expansion of wider 50mm designs of the MI-ECoG could be tested, demonstrating the application of the platform for larger ECoG designs (Fig. 2F).” Pg 5 paragraph 3

- SEEG probes require anchor-bolts through which the electrodes are threaded. This is why the electrode are kept at a same or smaller diameter compared to their cables. The authors should describe the mechanism by which they aim to insert and secure the significantly larger and planar structure.

We understand this comment to be referring to the flat flexible cable (FFC) that connects our MI-ECoG to an external recording system. FFCs are commonly used for connections to thin film neural interfacing electrodes and have been employed in human clinical studies for thin film ECoG devices¹⁻³. Similarly, for acute human clinical studies thin film ECoG devices, devices have been directly connected to headstage PCB without additional cabling for interfacing⁴⁻⁸. However we acknowledge for chronic application of these types of ECoGs, including the MI-ECoG presented in this work, further development on cabling and anchorable connections may be required.

In the future, a custom anchor bolt with an oblong slit matching the packaged implant and FFC profile could be employed in a similar fashion to SEEG anchor-bolts. Another possibility for future MI-ECoG implants employed for long term use will be to have a skull mounted recording/stimulation system that directly connects to the electrode array thereby minimizing the need for lengthy cables similar to the developments under clinical evaluation led by Neuralink and BioInduction. Reliable methodologies to attach thin-film devices to the cylindrical style cables used with conventional neural implants remains an active area of research and development in both industry and academia. Such development work is outside the scope of this manuscript however we anticipate use of cylindrical cabling will be possible in future iterations of the MI-ECoG platform.

We have expanded discussion about cabling and anchoring in the manuscript as follows:

“This will be the subject of future research alongside the integration of improved electrode connections that allow compatibility with surgical anchoring to the surrounding tissue when used for chronic recordings.” (Pg 11, paragraph 2)

-No data has been shown on the stability of the device. Such data is important even for short acute recordings.

As this is an acute test of our MI-ECoG, we do not have long-term electrical stability results available from chronic implantation studies. A chronic implantation study will be the focus of future research (the ethical approvals for which will be facilitated by publishing this acute study). As noted in the manuscript, we refer the reviewer and reader to a recent report⁹ investigating the stability of the electrode material system used in the MI-ECoG (ie. devices fabricated with the same materials and processes in the same facility albeit without the shape-actuation components which are independent of the electrodes). The effects of the mechanical stress incurred by shape-actuated folding and unfolding were directly investigated in this study. Specifically, we have included electrochemical characterisation of our PEDOT:PSS electrodes before and after packaging of the MI-ECoG, demonstrating electrode electrical stability to the packaging of the device for minimally invasive implantation. The EIS and CV characterisation of the MI-ECoG electrodes is included in Fig S3.

- A critical technical issue with intracranial devices is the reliable mechanical contact between the electrode and the surface of the brain. In small cranial windows there is often a significant flow of cerebrospinal fluid. What is the mechanism that maintains appropriate contact of the probe with the surface of the brain despite ongoing fluid flow? How do authors ensure the device is not detached and floating above the brain surface due to accumulating cerebrospinal fluid?

For the demonstration of the MI-ECoG, we used a 6mm slit incision in the dura instead of a full durotomy for implantation. This helps reduce both the swelling of the brain in the cranial window and helps to reduce CSF leakage, therefore helping to reduce excessive CSF flow that could cause floating of the device. As the device is fully implanted underneath the skull, the upwards pressure of brain onto the device post expansion also helps to ensure good cortical contact. As noted elsewhere in this letter, the quality of the subcranial recordings with the MI-ECoG were found to be at least as good as recordings from a clinical grid implanted post full craniotomy which indicates the device is appropriately contacting the surface of the brain.

For clarification, we have included the following statements into the manuscript

“The small incision in the dura helped to prevent unwanted swelling of the brain and reduced CSF leakage through the implantation site in comparison to a full durotomy.” (Pg5 , paragraph 1)

“To accurately test the expansion of the MI-ECoG, we ensured the device was implanted wholly underneath the skull using the custom tooling. This also helps ensure good cortical contact between the expanded MI-ECoG and the brain is achieved due to the intercranial pressure between the brain and the skull.” (Pg5 , paragraph 1)

- The manuscript is very light on actual electrophysiological data. Figure 3D is the only sample traces that can be used to evaluate the spatial resolution of the device. The recordings do not have a consistent gradient of the auditory response amplitude; instead, there is a sparse representation of the evoked response. This could be due to spatially inconsistent and poor contact between electrodes and the brain as mentioned above. It also seems that only two trials have been performed, and only one resulted in a response. In order for the authors to support their claims they should at least compare their device with the performance of a traditional ECoG array.

We thank the reviewer for this feedback which has helped improve our manuscript. We have now compared our device to a traditional ECoG array in a side-by-side in vivo experiment using the same animal (pig) and recording setup. We show the AEP spatial response recorded using the commercial ECoG device in Supplementary Figure 4 wherein a similar sparse response to AEPs than what was recorded using the MI-ECoG post-expansion is observed. We posit this reflects the fact that the ECoG electrodes are only partially covering the auditory cortex. When comparing the response of a MI-ECoG and commercial ECoG array implanted in the same porcine model in the updated Fig 3, it is evident the recordings of AEPs obtained by the MI-ECoG post-expansion had a similar amplitude response, and similar SNR profiles. The recordings obtained by both devices also had a similar power spectral density between 0.1-200Hz. This demonstrates that post-expansion, the MI-ECoG electrodes are in good contact to the cortical surface and can reliably obtain high quality electrophysiology recordings. To clarify, the MI-ECoG was successfully implanted and fully expanded with confirmation by both X-ray imaging and subsequent craniotomy in six pig trials. The first two trials were conducted in pigs immediately after euthanasia to confirm efficacy of expansion prior to in vivo tests and subsequently four in vivo trials were performed in anesthetized pigs.

For clarification, in the text we have included the following statements:

“A similar spatial response was observed from the PtIr ECoG (Supplementary Fig S4). The AEPs recorded by the MI-ECoG have similar profiles to previously described signals in anesthetised pigs, with a similar AEP response recorded by both the MI-ECoG post-expansion and the PtIr ECoG placed manually on the cortical surface (Fig 3F). The averaged power spectrum density of baseline and AEP recordings from both the MI-ECoG and PtIr shows minimal difference between the recordings from traditionally implanted PtIr grid and the MI-ECoG post-expansion (Fig 3G). The MI-ECoG also exhibited a similar signal-to-noise ratio (SNR) of 41 ± 1.8 dB compared to the SNR of 36.9 ± 2.5 dB for the PtIr when recording AEPs after expansion (Fig 3H). The slight increase in SNR in the MI-ECoG can be attributed to a slightly lower noise floor in the electrophysiological recordings by the device.” Pg 7, paragraph 4

“The recordings of AEPs obtained by the MI-ECoG were also similar to recording obtained using a traditionally implanted silicone, 200 μ m thick PtIr ECoG, with similar power density spectrum profiles and SNR to the recordings of AEPs obtained with the PtIr device. This demonstrates that the MI-ECoG had good contact with the cortical surface post-expansion and consequently is able to achieve a similar electrophysiological performance to a traditionally implanted array.” Pg 10, paragraph 2

Overall, this is an interesting concept and efforts in this direction are worthwhile, but the current manuscript requires extensive modification with additional experimentation and analysis before the authors' claims can be supported.

We thank the reviewer for their recognition of the value of this work and for their feedback to improve our manuscript.

References

1. Ganji, M. *et al.* Development and Translation of PEDOT:PSS Microelectrodes for Intraoperative Monitoring. *Adv. Funct. Mater.* **28**, 1–11 (2018).

2. Hermiz, J. *et al.* A clinic compatible, open source electrophysiology system. in *2016 38th Annual International Conference of the IEEE Engineering in Medicine and Biology Society (EMBC)* 4511–4514 (2016). doi:10.1109/EMBC.2016.7591730.
3. Hermiz, J. *et al.* Sub-millimeter ECoG pitch in human enables higher fidelity cognitive neural state estimation. *NeuroImage* **176**, 454–464 (2018).
4. Chiang, C. H. *et al.* Flexible, high-resolution thin-film electrodes for human and animal neural research. *J. Neural Eng.* **18**, 045009 (2021).
5. Hassan, A. R. *et al.* Translational Organic Neural Interface Devices at Single Neuron Resolution. *Adv. Sci.* **9**, 2202306 (2022).
6. Ho, E. *et al.* The Layer 7 Cortical Interface: A Scalable and Minimally Invasive Brain–Computer Interface Platform. 2022.01.02.474656 Preprint at <https://doi.org/10.1101/2022.01.02.474656> (2022).
7. Khodagholy, D. *et al.* Organic electronics for high-resolution electrocorticography of the human brain. *Sci. Adv.* **2**, e1601027 (2016).
8. Muller, L. *et al.* Thin-film, high-density micro-electrocorticographic decoding of a human cortical gyrus. *Conf. Proc. Annu. Int. Conf. IEEE Eng. Med. Biol. Soc. IEEE Eng. Med. Biol. Soc. Annu. Conf.* **2016**, 1528–1531 (2016).
9. Oldroyd, P. *et al.* Stability of Thin Film Neuromodulation Electrodes under Accelerated Aging Conditions. *Adv. Funct. Mater.* 2208881 (2022) doi:10.1002/ADFM.202208881.

REVIEWERS' COMMENTS

Reviewer #1 (Remarks to the Author):

The additional in vivo experiments and analysis on the electrophysiology data, and the emphasis on origami inspiration, make this paper sufficiently novel. I think the authors addressed all questions and comments properly in the revised manuscript. I now recommend this paper in Nature Communications.

I have one minor comment. The authors introduced previous studies on neural interfacing techniques for electrocorticography. I believe that additional references can support this part: e.g., "The ultra-thin, minimally invasive surface electrode array NeuroWeb for probing neural activity," Nature Communications 14, 7088 (2023), "Injectable ventral spinal stimulator evokes programmable and biomimetic hindlimb motion," Nano Letters 23, 6184-6192 (2023), and "Stitching flexible electronics into the brain," Advanced Science 10, 2300220 (2023).

Reviewer #2 (Remarks to the Author):

Authors were able to address my comments.

Reviewer #1 (Remarks to the Author):

The additional in vivo experiments and analysis on the electrophysiology data, and the emphasis on origami inspiration, make this paper sufficiently novel. I think the authors addressed all questions and comments properly in the revised manuscript. I now recommend this paper in Nature Communications.

I have one minor comment. The authors introduced previous studies on neural interfacing techniques for electrocorticography. I believe that additional references can support this part: e.g., "The ultra-thin, minimally invasive surface electrode array NeuroWeb for probing neural activity," Nature Communications 14, 7088 (2023), "Injectable ventral spinal stimulator evokes programmable and biomimetic hindlimb motion," Nano Letters 23, 6184-6192 (2023), and "Stitching flexible electronics into the brain," Advanced Science 10, 2300220 (2023).

Reviewer #2 (Remarks to the Author):

Authors were able to address my comments.

We thank the reviewers for their response to our revisions. As requested we have included the references into the following statements:

“Recent research has shown ECoGs can be made from softer^{16,17} and thinner^{18–22} materials to increase biocompatibility for a chronically implanted system, however, the invasiveness of implantation remains a key limitation to the wider adoption of these technologies.” (pg2, paragraph 1)

“Similarly, thin flexible mesh electrode arrays can be packaged within a microcatheter to allow introduction into the body via an injection^{25–28}.” (pg2, paragraph 2)